# Influence of transient pressure changes on speech intelligibility: Implications for next-generation train travel

Daniel Rooney[1]*, Martin Wittkowski[1], Susanne Bartels[1], Sarah Weidenfeld[1], Daniel Aeschbach[1,2,3]

1 Department of Sleep and Human Factors Research, Institute of Aerospace Medicine, German Aerospace Center (DLR), Cologne, Germany, 2 Division of Sleep and Circadian Disorders, Brigham and Women's Hospital, Boston, MA, United States of America, 3 Division of Sleep Medicine, Harvard Medical School, Boston, MA, United States of America

* daniel.rooney@dlr.de

**Data Availability Statement:** We are interested to share all data recorded in this study with anyone who wishes to access it for any purpose. We particularly believe external re-evaluation of our

## Abstract

High-speed trains are operated in increasingly complex railway networks and continual improvement of driver assistance systems is necessary to maintain safety. Speech offers the opportunity to provide information to the driver without disrupting visual attention. However, it is not known whether the transient pressure changes inside trains passing through tunnels interfere with speech intelligibility. Our primary goal was to test whether the most severe pressure variations occurring in high-speed trains (25 hPa in 2 s) affect speech intelligibility in individuals with normal hearing ability and secondly whether a potential effect would depend on the direction of the pressure change. A cross-over design was used to compare speech intelligibility, measured with the monosyllable word test by Wallenberg and Kollmeier, in steady ambient pressure versus subsequent to pressure events, both realised in a pressure chamber. Since data for a power calculation did not exist, we conducted a pilot study with 20 participants to estimate variance of intra-individual differences. The upper 80% confidence limit guided sample size of the main campaign, which was performed with 72 participants to identify a 10% difference while limiting alpha (5%) and beta error (10%). On average, a participant understood 0.7 fewer words following a pressure change event compared to listening in steady ambient pressure. However, this intra-individual differences varied strongly between participants, standard deviation (SD) ± 4.5 words, resulting in a negligible effect size of 0.1 and the Wilcoxon signed rank test (Z = -1.26; p = 0.21) did not distinguish it from chance. When comparing decreasing and increasing pressure events an average of 0.2 fewer words were understood (± 3.9 SD). The most severe pressure changes expected to occur in high-speed trains passing through tunnels do not interfere with speech intelligibility and are in itself not a risk factor for loss of verbal information transmission.

analysis is very valuable. All relevant data are within the manuscript and its Supporting Information files.

**Funding:** This study was part of the Next Generation Train project and funded by the Program Directorate Transport of the German Aerospace Center (www.dlr.de). The funder had no role in study design, data collection and analysis, decision to publish, or preparation of the manuscript.

**Competing interests:** Daniel Rooney, Martin Wittkowski, Susanne Bartels, Sarah Weidenfeld and Daniel Aeschbach have declared that no competing interests exist.

# 1 Introduction

Modern societies are characterised by a growing demand for mobility of individuals and high-speed trains are increasingly becoming a backbone of the transportation infrastructure in many countries [1]. To facilitate safe operation of these fast trains in increasingly complex railway networks a multitude of driver assistance systems are currently under development [2]. In human-machine interfaces the auditory channel can be used to provide information to the user, without disrupting visual attention [3]. This is particularly advantageous when operating heavy mobile machinery, such as trains. Speech has hereby an advantage over other acoustic stimuli; it does not only alert the operator but it carries the relevant information itself. However, if speech is considered for transmission of essential information in an operational environment it must be ensured that the acoustical transmission path is interruption-free. Therefore it is necessary to assess the train environment for interference with speech intelligibility.

Clearly, speech can only be understood, if it is heard. A basic metric used to describe audibility is therefore the sound pressure level (SPL) of the speech signal relative to the background noise, i.e. signal-to-noise ratio (SNR). A rule of thumb suggests an SNR of 6 dB(A) is required for minimal and 10 dB(A) for good speech intelligibility [4]. Beyond SPL, spectral composition of speech and noise also play an important role [5], e.g. the more noise intersperses into the speech frequency band the stronger its disruptive impact. More advanced metrics, such as the Speech Transmission Index [6], the Articulation Index [7] and the Speech Intelligibility Index [8] incorporate additional acoustic properties to deduce transmission quality and the amount of speech reaching the listener. But while the influence of acoustic parameters is well understood, little attention has been paid to other environmental factors, such as ambient pressure, humidity or temperature [9].

The fast transient pressure changes experienced in high-speed trains when passing through tunnels [10] are associated with aural discomfort [11], can attenuate sound transmission through the outer and middle ear [12] and have thus the potential to interfere with speech intelligibility. Repeated tunnel passing has even been linked to persistent hearing loss in train drivers [13], though the primary concern here was tunnel noise. In general, negative middle ear pressure is a common pathological dysfunction and has been intensely studied [14]. It is known that static pressure gradients across the tympanic membrane impair hearing sensitivity [15] and speech intelligibility has been looked at as a function of altitude in aircraft [16]. However, the long and slow pressure gradients in airplanes provide opportunity to equilibrate middle ear pressure and induce hypobaric hypoxia with increasing altitude. To our knowledge, no data is available to discern whether the small and rapid transient pressure variations occurring inside high-speed trains passing through tunnels can affect intelligibility of speech. Consequently, it is not known whether such variations may be a source of interference when using speech to indicate potentially critical information to train drivers.

Our primary objective was to test whether transient pressure changes of 25 hPa in 2 s, the most severe pressure events expected to occur in high-speed trains [10], affect speech intelligibility in individuals with normal hearing ability. Our secondary question was to assess whether the direction of the pressure change, i.e. increasing or decreasing ambient pressure, affects understanding of speech differently. We used the monosyllabic rhyme test devised by Wallenberg and Kollmeier (WAKO) to measure speech intelligibility [17], which is based on understanding of individual words.

## 2 Material and methods

This prospective study was approved by the ethical committee of the medical association North Rhine-Westphalia (approval date June 6, 2017; number 2017121) and performed at the German Aerospace Centre (DLR) in Cologne (Germany). The pilot study for sample size estimation was performed in August 2017 and the main data collection took place between September and October 2017.

### 2.1 Sample size estimation

It was our aim to conclusively answer the question as to whether or not speech intelligibility is affected by transient pressure changes. Since statistical tests only control for false positive results it was crucial to study a sufficiently sized sample in order to restrict false negative findings as well. We could not identify any published data enabling a reliable power calculation; therefore we conducted a pilot study with 20 participants (10 female, mean age 27 years ± 6 SD). Variability estimates from pilot data tend to underestimate the population parameter [18]. To account for this we calculated the upper 80% confidence limit for the standard deviation (SD) of intra-individual differences (7.45 words) and used it to guide sample size of the main campaign [19]. This resulted in recruitment of 72 participants, enabling us to identify a 10% difference in speech intelligibility within an individual, while limiting the chance of alpha and beta error to 5% and 10% respectively. Rounding the sample size up to the next multiple of 8 was required to enable permutation of experimental conditions.

### 2.2 Participants

Both female and male individuals with unimpaired hearing ability were eligible for the experiment. Out of the 72 volunteers, aged 19 to 39 years (mean 25 ± 4 SD), participating in the main campaign 35 were female. Pilot study participants were not included in the main trial. Participants were recruited from a database of former study attenders at the DLR Institute of Aerospace Medicine and spoke German as their native language. Normal hearing ability, defined as no more than 20 dB reduction in any frequency band of 500 Hz and 1000 to 4000 Hz (in 1000 Hz steps), middle ear pressure and mobility of the eardrum were ensured by audiometry (Device AD226, Interacoustics, Audiometer Alle, 5500 Middelfart, Denmark) and tympanometry (Device TITAN, also Interacoustics), both measured before and after the experimental procedure in the pressure chamber. All volunteers gave written informed consent before starting the test protocol and were in good health as assessed by general questionnaires. Participation was reimbursed with 90 Euro.

### 2.3 Word test

Speech intelligibility was measured using the monosyllable word test by Wallenberg and Kollmeier (WAKO) [17], commercially distributed by HörTech gGmbH, Oldenburg, Germany. The test uses a closed-set response method; subsequent to each audibly presented test word the participant is asked to identify the word from five written alternatives. Each alternative differs in one of three phonemes. The test can be used in quiet and in noise and since it is based on the understanding of individual words it qualifies to discern the effect of short recurrent events, like changes in ambient pressure.

### 2.4 Experimental design

The pressure chamber used for the experiment is a former deep diving facility, essentially a horizontally oriented metal cylinder, capable of producing rapid and highly accurate pressure

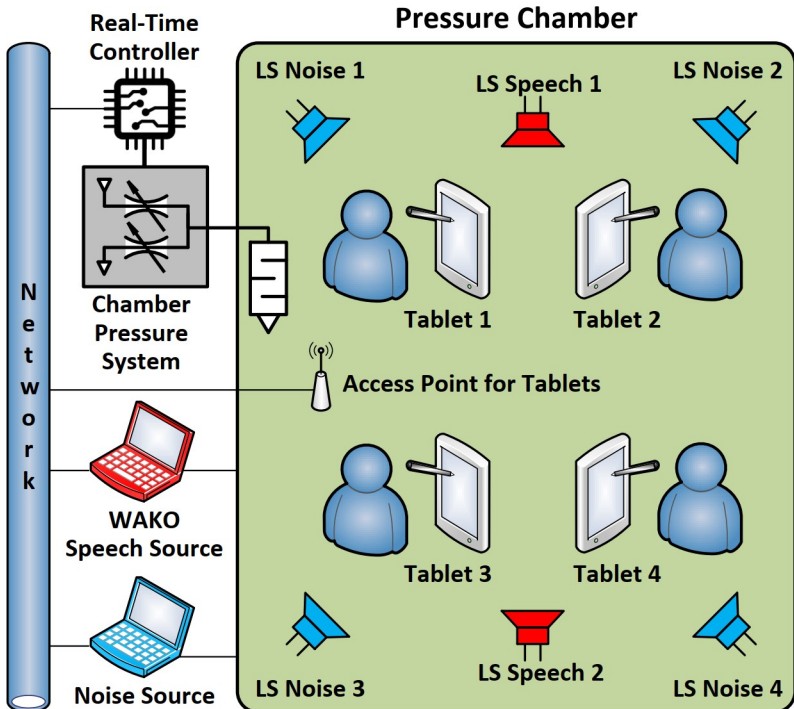

**Fig 1. Experimental assembly in the pressure chamber.** A real time controller synchronised chamber pressure, announcement of WAKO test words and response collection via tablet computers. A homogeneous acoustic environment was created using independent signal processors for each of the four loudspeakers (LS) for noise and the two LS for test word playback.

variations [11]. The barrel-shape of the facility is well suited to create homogeneous noise environments and has been used for this purpose in previous studies simulating train environments [20]. We used four loudspeakers for playback of the background noise, two at both ends of the cylinder, each facing the round chamber wall. In addition, two separate loudspeakers, installed at the round ceiling, were used for announcement of the test words. The sound at each seat was calibrated using a microphone at ear height. A schematic of the test set-up is shown in Fig 1.

We used a cross-over design in which each participant was studied in a single 1 hour session. Participants were examined in groups of four and each session started at 1 o'clock in the afternoon with a familiarisation to the speech intelligibility task using the same 10 test words for all participants. Afterwards, participants were presented with two test blocks of 50 words each. In one block, each test word was played without any delay after a linear change in ambient pressure of 25 hPa within 2 s, while the other block was performed at steady ambient pressure of 950 hPa. To account for potential order effects (e.g. due to time on task), we permuted the experimental conditions, i.e. testing with and without pressure changes, while maintaining the order of words throughout each test block. To call attention to the upcoming test word each presentation was preceded by an audible announcement ("please mark the word"). Each word was followed by a 20 second break during which participants selected the answer on a tablet computer and had the opportunity to equilibrate middle ear pressure. Instead of repetitively presenting pairs of increasing and decreasing pressure we constructed an irregular sequence of positive and negative pressure changes. This prevented participants from anticipating the direction of the upcoming pressure change. The sequence was balanced in regard to increasing and decreasing events and the number of successions in each direction. The

minimum chamber pressure was restricted to 900 hPa in order to avoid onset of hypoxia. A speech simulating background noise (recommended spectral composition for WAKO) of 67 dB(A) was constantly played throughout the experiment, simulating the acoustic surrounding of a train and masking the operating sounds of the chamber. Prior to our experiment the direction of the effect was unknown, i.e. it was not clear whether pressure changes would increase or decrease speech intelligibility and it was possible that pressure changes in different directions could have opposite effects. Thus, the experiment needed to be equally sensitive in both directions. This was achieved by calibrating SPL of the presented test words to yield 50% speech intelligibility in steady ambient pressure, resulting in an SNR of approximately 0 dB, i.e. speech and noise signal had nearly the same SPL.

## 2.5 Data analysis

R version 3.6.1 (R Foundation for Statistical Computing, Vienna, Austria) was used to perform all calculations. We planned to test both our hypotheses using a two-sided Wilcoxon signed-rank test with Pratts extension to handle ties [21], since we expected the data to be symmetrically but not normally distributed. To maintain an overall family-wise error rate of 5% we devised hierarchical testing using complete alpha spending from the primary to the secondary hypothesis, i.e. only testing for an effect of direction if we observed an overall effect of pressure changes on the number of correctly identified words within participants. After we had defined our a priori analysis protocol a simulation study of paired count data was published [22], attesting that we had chosen an appropriate and unbiased strategy, but that a paired t-test could, in comparison, yield slightly improved power, despite the data not being compliant with its test assumptions. This encouraged us to present standard deviations and t-distribution based confidence intervals of our results.

Since we expected missing data to occur unsystematically and only occasionally as a result of attention lapses, we did not plan an imputation strategy. Participants would immediately be notified in case they missed a response, bringing their attention back to the task. We recorded 7194 of the nominal 7200 responses, the six missing replies originated from five different participants and three occurred in each experimental condition. The assumption of these being missing completely at random appears justified.

## 3 Results

In the condition without pressure variations the average number of correctly identified words per participant was 29.9 (± 3.4 SD) out of 50, i.e. 49.7% (± 8.5 SD) of words were truly understood when considering each trial had a 20% chance for correct guessing. This shows that the targeted calibration of a 50% intelligibility rate at steady ambient pressure was achieved within reason.

When comparing the number of correctly understood test words announced immediately following a pressure change event to the number correctly understood in steady ambient pressure within each participant, the average understanding was 0.7 fewer words in the former condition. Due to the high variability of this intra-individual differences between the study participants, it has a standard deviation (SD) of ± 4.5 words, this equates only to a negligible effect size of 0.1 and the Wilcoxon signed rank test (Z = -1.26; p = 0.21) does not detect it to be distinguishable from chance level. When comparing decreasing and increasing pressure events within each participant the average understanding differed by 0.2 words (± 3.9 SD), but this was not formally tested due to our hierarchically ordered hypothesis and lack of statistical significance of the primary hypothesis. The corresponding confidence intervals are displayed in Fig 2. Visibly, no difference, i.e. zero, is within the interval borders.

## 4 Discussion

Disequilibrium of tympanic air pressure has been reported to promote conductive hearing loss by changing the mechanical properties of the tympanum [23]. Thus, it has not been farfetched to consider pressure variations in trains passing through tunnels a risk factor for verbal information loss. To close this knowledge gap we studied the phenomenon using a highly controlled experimental setup and examined both positive and negative variations in ambient pressure. The results of this study indicate that speech intelligibility is not impaired by transient pressure changes in the order of magnitude expected during train travel.

We studied four participants during each test session and used an established calibration procedure to assure the quality of the acoustic environment [20]. But since the calibration was performed without having participants, i.e. four human bodies, inside the chamber, it is still possible that the acoustic properties at the different seats may have varied slightly during the actual experiments. However, the within-subject design, in combination with the fact that participants did not change seats during the procedure, ensures that the intra-individual differences in words understood are an unbiased representation of the effect of pressure changes on speech intelligibility.

We chose amplitude (25 hPa) and duration (2 s) of our intervention to slightly exceed the severity of events that occupants experience aboard current high-speed trains [10]. The positive dose-response relationship between aural impairment and severity of pressure events [15] enables to generalise our findings to scenarios of lesser severity, essentially covering the complete range of expectable pressure events in trains. Due to the level of discomfort associated with pressure variations it is very unlikely that current limits for such events in trains will change in the future [11]. Since we considered it equally important to quantify our confidence in positive as well as negative findings, we paid close attention to selecting an appropriately sized sample [18]. This secures the validity of our negative result, i.e. no difference between the two experimental conditions, with 90% certainty.

Prior studies have shown changes in aural perception by negative middle ear pressure, particularly in the speech band [15]. This is considered to have a degrading effect on the understanding of speech and is known to impair language acquisition in children with this condition [24]. It has been hypothesised that positive pressure in the ear canal would cause analogous effects [14]. At first glance, this appears contradictory to our observation of unimpaired speech intelligibility subsequent to bidirectional pressure variations, but an important difference must be taken into account: while previous physiological studies were carried out in quiet, our study measured understanding of speech in the presence of background noise, similar to what is experienced during actual train travel [20]. From a mechanistic point of view, pressure gradients across the tympanic membrane will lead to changes in the transfer function of the hearing organ [25], i.e. frequencies across the hearing spectrum are differently attenuated. This may reduce the power of relevant components in the speech signal, which in turn reduces its intelligibility [5]. However, when listening to speech at sufficient SPL in the presence of background noise the main problem for understanding becomes the overlay of the polluting signal [4]. In the noisy train environment any attenuation due to un-equilibrated ear pressure will, assuming a similar frequency composition, affect the speech and the noise signal equally and consequently not change the overall SNR. The test words in the experiment were announced with approximately the same SPL as the background noise playback, i.e. 67 dB(A). It appears reasonable to assume that, despite frequency attenuation due to ear pressure, all characteristics relevant for speech intelligibility will have remained well above the hearing threshold. In essence, we speculate that we did not observe a difference in speech intelligibility between our experimental conditions since the pressure events did not significantly change

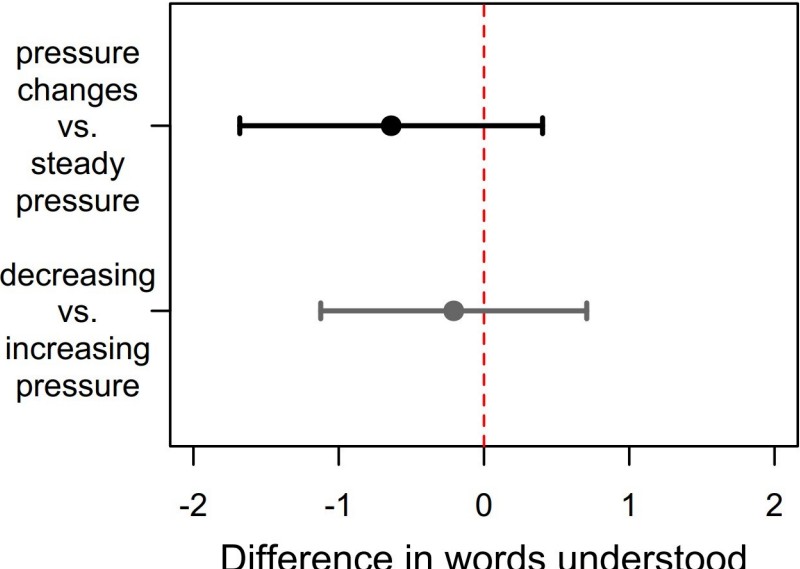

**Fig 2. Estimators and 95%-confidence intervals.** Intra-individual differences in number of words understood, CI based on t-distribution (71 degrees of freedom).

SNR or audibility of the speech signal, giving participants in both conditions approximately 50% chance to understand the test words.

While our results imply that pressure variations in trains caused by tunnels are unproblematic for using speech as a communication modality in driver assistance systems, the tunnel noise may still pose a considerable obstacle for this application. For good speech intelligibility a suitable SNR, approximately 10 dB(A) [4], must be ensured under any circumstances but at the same time peak volume of announcements must be limited to prevent noise stress and its detrimental short and long term consequences [26]. Thus, only if the operational environment is sufficiently quieted, e.g. by proper insulation/sound-attenuation of the vehicle body, speech based information transmission will be a safe tool.

## 5 Conclusion

High-speed train drivers are not only operating their vehicles in ever more complex railway infrastructures, but they are also subject to a multitude of factors, such as monotonous operating cycles and shift work, known to increase the likelihood of fatigue and human error [27], consequences of which may be catastrophic. Advanced assistance systems are a necessity to support train drivers and ensure safe rail operation in this ever more demanding environment. The present study found that fast transient pressure changes in itself did not interfere with speech intelligibility and therefore do not appear to be a risk factor for disruption of verbal communication. This suggests that driver assistance systems can safely make use of speech to communicate relevant information. However, speech intelligibility requires sufficient SNR and a limitation of the overall sound pressure level. The train driver must therefore operate in a sufficiently quiet environment. While this may be achieved by acoustic-insulation of the train body, it is highly desirable to aim for an overall reduction of noise emission by the train, since railway noise is not only a relevant issue inside the cabin, but also an overarching problem for residents in proximity of railway lines [28], causing annoyance and sleep disturbance [29].

## Supporting information

**S1 File.**
(PDF)

## Acknowledgments

We are grateful to all volunteers who made this research possible by contributing their time. We thank Jana-Kosima Schwarzlos-Sooprayen for helping with the selection of the speech intelligibly test and Julia Quehl for supporting the preparation of the experiment. HörTech gGmbH provided us with digital versions of the test word lists, which we sincerely appreciate.

## Author Contributions

**Conceptualization:** Daniel Rooney, Martin Wittkowski, Susanne Bartels, Daniel Aeschbach.

**Data curation:** Daniel Rooney.

**Formal analysis:** Daniel Rooney, Martin Wittkowski, Daniel Aeschbach.

**Funding acquisition:** Daniel Rooney, Daniel Aeschbach.

**Investigation:** Daniel Rooney, Martin Wittkowski, Sarah Weidenfeld.

**Methodology:** Daniel Rooney, Susanne Bartels.

**Project administration:** Daniel Rooney, Martin Wittkowski.

**Supervision:** Daniel Rooney, Daniel Aeschbach.

**Validation:** Daniel Rooney, Susanne Bartels, Sarah Weidenfeld, Daniel Aeschbach.

**Visualization:** Daniel Rooney.

**Writing – original draft:** Daniel Rooney.

**Writing – review & editing:** Daniel Rooney, Martin Wittkowski, Susanne Bartels, Sarah Weidenfeld, Daniel Aeschbach.

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
