## [Decision Letter · Decision Letter 0]

13 Feb 2020

PONE-D-19-26567

Influence of transient pressure changes on speech intelligibility: implications for next-generation train travel

PLOS ONE

Dear Mr. Rooney,

The manuscript has been assessed by two reviewers; their comments are available below.

While one of the reviewers has provided a positive assessment, the other has noted major concerns about aspects of the study design and notes that further experimental data are required to support the conclusions drawn.

We ask that you revise the manuscript to carefully address the points raised during the review process.

Please note that the revised manuscript will need to undergo further review, we thus cannot at this point anticipate the outcome of the evaluation process.

We would appreciate receiving your revised manuscript by Mar 29 2020 11:59PM. Please include the following items when submitting your revised manuscript:

We look forward to receiving your revised manuscript.

Kind regards,

Iratxe Puebla

Deputy Editor-in-Chief, PLOS ONE

Journal Requirements:

Reviewers' comments:

Reviewer's Responses to Questions

**Comments to the Author**

1. Is the manuscript technically sound, and do the data support the conclusions?

Reviewer #1: No

Reviewer #2: Yes

2. Has the statistical analysis been performed appropriately and rigorously? 

Reviewer #1: Yes

Reviewer #2: Yes

3. Have the authors made all data underlying the findings in their manuscript fully available?

Reviewer #1: Yes

Reviewer #2: Yes

4. Is the manuscript presented in an intelligible fashion and written in standard English?

Reviewer #1: No

Reviewer #2: Yes

5. Review Comments to the Author

Reviewer #1: This study examined the effects of transient changes in atmospheric pressure on speech intelligibility in travel trains. There has been little research on the effects of environmental factors such as ambient pressure on speech intelligibility. Therefore, the originality of this study is very high and I think it is an interesting research topic.

However, the scope of the experiment designed in this study is very limited, and complementary is essential for the experimental methodology. A study by Bostron et al. (2011) found that atmospheric effects in airplanes are small at low or high SNR but significant at moderate SNR. However, this study only evaluates speech intelligibility for the same sound pressure level (SNR = 0 dB) as background noise, and therefore has a clear limitation in the experimental design. In addition, although the direction of the speaker that reproduces noise in the experimental environment is toward the subject, it is considered that a sufficient diffused sound field will be realized only when the direction of the speaker faces the wall. If not, the speaker setup used by the author needs to be compensated for because of the directional influence of the sound source.

Therefore, this paper is not suitable for publication on PLOS ONE.

[9] Bostron JH, Brungart TA, Barnard AR, McDevitt TE. Atmospheric effects on voice command intelligibility from acoustic hail and warning devices. The Journal of the Acoustical Society of America. 2011;129(4):2237-44. Epub 2011/04/12. doi: 10.1121/1.3559710. PubMed PMID: 21476678.

Reviewer #2: The paper report a study about intelligibility in trains due to changes in pressure, performed by means of measurements and questionnaires. The study is correctly performed, but no significant results are obtained. This will not be a paper that can leave the mark, but is still publishable.

Minor correction are reported to the authors.

• I suggest to avoid the abstract style used, which is too analytic. It is important to report what are the background, objective, methods, and so on, but it is not important to read them explicitly written.

• A result of 0.7 plus or less 4.5 is a very bad number in math. It is simply very bad to read and should be reported in some different ways.

• Conclusions are a bit short and should summarize better the paper. Furthermore, I suggest, and kindly ask, to mention at the end a sentence giving attention to the very important topic of railway noise impact over citizens, which is still very annoying and deserving attention and prevention. “However, railway noise still remains an issue for people living in the nearby of railway axis, deserving community attention and careful planning in order to avoid citizens’ complaints and annoyance [Licitra, G., Fredianelli, L., Petri, D., & Vigotti, M. A. (2016). Annoyance evaluation due to overall railway noise and vibration in Pisa urban areas. Science of the total environment, 568, 1315-1325; Bunn, F., & Zannin, P. H. T. (2016). Assessment of railway noise in an urban setting. Applied Acoustics, 104, 16-23.].

6. PLOS authors have the option to publish the peer review history of their article (what does this mean?). If published, this will include your full peer review and any attached files.

Reviewer #1: No

Reviewer #2: No

---

## [Author Response · Author response to Decision Letter 0]

29 Mar 2020

Answers to comments from Reviewer 1

This study examined the effects of transient changes in atmospheric pressure on speech intelligibility in travel trains. There has been little research on the effects of environmental factors such as ambient pressure on speech intelligibility. Therefore, the originality of this study is very high and I think it is an interesting research topic. 

Question 1: However, the scope of the experiment designed in this study is very limited, and complementary is essential for the experimental methodology. A study by Bostron et al. (2011) found that atmospheric effects in airplanes are small at low or high SNR but significant at moderate SNR. However, this study only evaluates speech intelligibility for the same sound pressure level (SNR = 0 dB) as background noise, and therefore has a clear limitation in the experimental design. 

The study of Bostron et al. examines propagation of sound in different atmospheric conditions, i.e. combinations of pressure, humidity and temperature. We cited the afore-mentioned study in our initial manuscript and have now stated their findings more explicit in the methods section, since these are crucial for the experimental setup we chose. Bostron et al. studied intelligibility of voice commands from acoustic warning devices, travelling over different distances on an airfield (receivers were positioned in 7 locations between 200m and 1500m away from the sender) at different days (i.e. different sets of atmospheric parameters, determined by the meteorological conditions of that day). The signal to noise ratio (SNR) at the different receiver positions was one of the explanatory variables they measured. They found that sound pressure levels (SPL) did not only decrease with distance, but were also affected by the meteorological conditions. They studied different SNR conditions by varying SPL of the voice commands and found that atmospheric conditions did only affect intelligibility of the voice commands at moderate SNR. At both high and low SNR, intelligibility of the voice commands was nearly independent of the atmospheric conditions, i.e. in the former voice commands were always understood and in the latter they were never understood.

Since we intended to study the isolated effect of pressure changes on speech intelligibility (not the effect of different absolute pressure levels as Bostron et al. did) this observation is pivotal for our study design and it is the particular reason why we exclusively measured at SNR = 0 dB. Prior to our experiment the direction of the effect of transient pressure changes was unknown, i.e. it was not clear whether pressure changes would increase or decrease speech intelligibility and it was also possible that pressure changes in different directions (i.e. increasing or decreasing pressure) could have opposite effects. Thus, our experiment needed to be equally sensitive in both directions. When the monosyllable word test by Wallenberg and Kollmeier is done at 0 dB SNR it provides a 50% chance for correct understanding of the test words, i.e. participants should correctly identify half of the test words (after removing the 1 in 5 chance for correct guessing). We checked the realisation of this assumption post-hoc by looking at the results of the steady ambient pressure condition only and found 49.7% of test words were identified correctly. By having the test in this “middle position”, our experiment was equally sensitive for increases as well as decreases in speech intelligibility and this was what we aimed for. Studying other SNR conditions would have provided information on the effect of SNR on speech intelligibility, which was not the goal of our experiment. Adding additional SNR scenarios would have increased the length of the experimental procedure beyond the attention span known to be acceptable for participants in this type of experiment and it would have increased the likelihood of floor and/or ceiling effects. To avoid these two problems we studied exclusively at an SNR of 0 dB, since this was optimal in the scope of our experiment.

To illustrate this situation we added the following sentence to the methods section:

Prior to our experiment the direction of the effect was unknown, i.e. it was not clear whether pressure changes would increase or decrease speech intelligibility and it was possible that pressure changes in different directions could have opposite effects. Thus, the experiment needed to be equally sensitive in both directions. This was achieved by calibrating SPL of the presented test words to yield 50% speech intelligibility in steady ambient pressure, resulting in an SNR of approximately 0 dB, i.e. speech and noise signal had nearly the same SPL.

[9] Bostron JH, Brungart TA, Barnard AR, McDevitt TE. Atmospheric effects on voice command intelligibility from acoustic hail and warning devices. The Journal of the Acoustical Society of America. 2011;129(4):2237-44. Epub 2011/04/12. doi: 10.1121/1.3559710. PubMed PMID: 21476678.

The indicated scenario of speech intelligibility as a function of altitude, i.e. pressure, in airplanes was studied by Wagstaff et al. (1999). However, the slow and gradual pressure gradients in airplanes do not compare to the rapid pressure variations experienced in the train scenario studied here. The former provides opportunity to equilibrate in ear pressure and induces hypobaric hypoxia with increasing altitude, while the latter scenario prevents pressure equilibration due to the brevity of the events and does not induce hypoxia. Wagstaff et al. looked at the effects of different constants pressure levels on hearing, while the novelty of our study is that it investigates the immediate effect of small but rapid pressure events (without inducing hypoxia) on speech intelligibility. 

We added sentences to this effect to the introduction:

It is known that static pressure gradients across the tympanic membrane impair hearing sensitivity [15] and speech intelligibility has been looked at as a function of altitude in aircraft [16]. However, the long and slow pressure gradients in airplanes provide opportunity to equilibrate middle ear pressure and induce hypobaric hypoxia with increasing altitude. To our knowledge, no data is available to discern whether the small and rapid transient pressure variations occurring inside high-speed trains passing through tunnels can affect intelligibility of speech.

[16] Wagstaff AS, Tvete O, Ludvigsen B. Speech intelligibility in aircraft noise as a function of altitude. Aviation, space, and environmental medicine. 1999;70(11):1064-9. Epub 1999/12/23. PubMed PMID: 10608602.

Question 2: In addition, although the direction of the speaker that reproduces noise in the experimental environment is toward the subject, it is considered that a sufficient diffused sound field will be realized only when the direction of the speaker faces the wall. If not, the speaker setup used by the author needs to be compensated for because of the directional influence of the sound source.

Figure 1 in the manuscript represents only a schematic of the experimental assembly and the depicted orientation of loudspeakers serves illustrative purposes; it is different from the actual orientation of the loudspeakers in the test facility. The pressure chamber used in this study is a former deep diving facility; it is essentially a horizontally oriented metal cylinder (see Photo P1 and P2 below). This barrel-shape of the facility is well suited to create homogeneous noise environments, which we have already done in previous studies simulating train and airplane environments (e.g. Sanok et al. 2015). We used four loudspeakers for playback of the background noise, two at both ends of the cylinder, each facing the round chamber wall. In addition, two separate loudspeakers were used for announcement of the test words. These two were installed face-up at the top “ceiling” of the round cylinder. Each loudspeaker was controlled by an individual DSP, providing complete control over the sound environment inside the chamber. Prior to the experiment the sound at each seat was specifically fine-tuned, using a calibration microphone at ear height. Due to this established procedure we have confidence in the quality of the acoustic environment realised during the experiment. However, even if acoustic differences between seats would have existed, this would not invalidate the experimental results. Due to the within-subject design and the fact that participants did not change seats during the study procedure the intra-individual differences between the pressure-change and no-pressure-change condition would still be a valid representation of the effect of this variable.

We added detailing sentences to Material and Methods:

The pressure chamber used for the experiment is a former deep diving facility, essentially a horizontally oriented metal cylinder, capable of producing rapid and highly accurate pressure variations [11]. The barrel-shape of the facility is well suited to create homogeneous noise environments and has been used for this purpose in previous studies simulating train environments [20]. We used four loudspeakers for playback of the background noise, two at both ends of the cylinder, each facing the round chamber wall. In addition, two separate loudspeakers, installed at the round ceiling, were used for announcement of the test words. The sound at each seat was calibrated using a microphone at ear height. 

and Discussion:

We studied four participants during each test session and used an established calibration procedure to assure the quality of the acoustic environment [20]. But since the calibration was performed without having participants, i.e. four human bodies, inside the chamber, it is still possible that the acoustic properties at the different seats may have varied slightly during the actual experiments. However, the within-subject design, in combination with the fact that participants did not change seats during the procedure, ensures that the intra-individual differences in words understood are an unbiased representation of the effect of pressure changes on speech intelligibility.

[20] Sanok S, Mendolia F, Wittkowski M, Rooney D, Putzke M, Aeschbach D. Passenger comfort on high-speed trains: effect of tunnel noise on the subjective assessment of pressure variations. Ergonomics. 2015;58(6):1022-31. Epub 2015/01/20. doi: 10.1080/00140139.2014.997805. PubMed PMID: 25597694.

Photo P1: interior view of pressure chamber Photo P2: exterior view of pressure chamber

We hope our explanations show that we have paid great attention to every detail of our experiment, in order to make the design as well as the implementation of the study as scientifically sound as possible. We therefore respectfully disagree with the notion that, due to a lack of rigour in our experimental procedure, our manuscript was not suitable for publication in PLOS ONE.

Answers to comments from Reviewer 2

The paper report a study about intelligibility in trains due to changes in pressure, performed by means of measurements and questionnaires. The study is correctly performed, but no significant results are obtained. This will not be a paper that can leave the mark, but is still publishable.

Minor correction are reported to the authors:

Question 1: I suggest to avoid the abstract style used, which is too analytic. It is important to report what are the background, objective, methods, and so on, but it is not important to read them explicitly written.

We removed all structural-words from the abstract to make it more readable. 

Please see Abstract in the revised version of the manuscript.

Question 2: A result of 0.7 plus or less 4.5 is a very bad number in math. It is simply very bad to read and should be reported in some different ways.

We agree, it may feel funny to read of 0.7 ± 4.5 words, since 0.7 words do not make sense in the real world. And since we use the Wilcoxon signed rank test we thought initially of reporting the intra-individual differences by median, as measure of central tendency, and quartiles to describe variability. However, Proudfoot et al. (2018) showed that describing paired count data by mean and standard deviation is a valid approach and this parametric representation has the advantage that it speaks naturally to the “internal statistics processor” of most people, i.e. the interpretation of mean = 0.7 words ± 4.5 SD as a non-significant result will be self-evident for most people, while a non-parametric representation of the data would not be. We are reporting a negative result of a sufficiently powered study, unfortunately still not a “familiar thing” for many researchers, and think it is therefore important for our results to be unambiguous and easy to interpret. For this reason we would like to keep the valid but odd-looking parametric representation we chose in the initial manuscript. However, we reworded all passages describing these numbers to improve readability and to make the underlying arithmetics very clear:

In Abstract:

On average, a participant understood 0.7 fewer words following a pressure change event compared to listening in steady ambient pressure. However, this intra-individual differences varied strongly between participants, standard deviation (SD) ± 4.5 words, resulting in a negligible effect size of 0.1 and the Wilcoxon signed rank test (Z=-1.26; p=0.21) did not distinguish it from chance. When comparing decreasing and increasing pressure events an average of 0.2 fewer words were understood (± 3.9 SD).

In Results:

When comparing the number of correctly understood test words announced immediately following a pressure change event to the number correctly understood in steady ambient pressure within each participant, the average understanding was 0.7 fewer words in the former condition. Due to the high variability of this intra-individual differences between the study participants, it has a standard deviation (SD) of ± 4.5 words, this equates only to a negligible effect size of 0.1 and the Wilcoxon signed rank test (Z=-1.26; p=0.21) does not detect it to be distinguishable from chance level. When comparing decreasing and increasing pressure events within each participant the average understanding differed by 0.2 words (± 3.9 SD), but this was not formally tested due to our hierarchically ordered hypothesis and lack of statistical significance of the primary hypothesis.

[22] Proudfoot JA, Lin T, Wang B, Tu XM. Tests for paired count outcomes. General psychiatry. 2018;31(1):e100004. Epub 2018/12/26. doi: 10.1136/gpsych-2018-100004. PubMed PMID: 30582120; PubMed Central PMCID: PMCPMC6211281.

Question 3: Conclusions are a bit short and should summarize better the paper. Furthermore, I suggest, and kindly ask, to mention at the end a sentence giving attention to the very important topic of railway noise impact over citizens, which is still very annoying and deserving attention and prevention. “However, railway noise still remains an issue for people living in the nearby of railway axis, deserving community attention and careful planning in order to avoid citizens’ complaints and annoyance [Licitra, G., Fredianelli, L., Petri, D., & Vigotti, M. A. (2016). Annoyance evaluation due to overall railway noise and vibration in Pisa urban areas. Science of the total environment, 568, 1315-1325; Bunn, F., & Zannin, P. H. T. (2016). Assessment of railway noise in an urban setting. Applied Acoustics, 104, 16-23.].

We extended the conclusions section in the manuscript to provide a more complete summary of our findings. We also emphasised that our experiment only addressed the isolated effect of pressure changes inside of trains and point to the general issue of railway noise:

The present study found that fast transient pressure changes in itself did not interfere with speech intelligibility and therefore do not appear to be a risk factor for disruption of verbal communication. This suggests that driver assistance systems can safely make use of speech to communicate relevant information. However, speech intelligibility requires sufficient SNR and a limitation of the overall sound pressure level. The train driver must therefore operate in a sufficiently quiet environment. While this may be achieved by acoustic-insulation of the train body, it is highly desirable to aim for an overall reduction of noise emission by the train, since railway noise is not only a relevant issue inside the cabin, but also an overarching problem for residents in proximity of railway lines [28], causing annoyance and sleep disturbance [29].

[28] Licitra G, Fredianelli L, Petri D, Vigotti MA. Annoyance evaluation due to overall railway noise and vibration in Pisa urban areas. The Science of the total environment. 2016;568:1315-25. Epub 2016/01/19. doi: 10.1016/j.scitotenv.2015.11.071. PubMed PMID: 26775834.

[29] Pennig S, Quehl J, Mueller U, Rolny V, Maass H, Basner M, et al. Annoyance and self-reported sleep disturbance due to night-time railway noise examined in the field. The Journal of the Acoustical Society of America. 2012;132(5):3109-17. Epub 2012/11/14. doi: 10.1121/1.4757732. PubMed PMID: 23145596.

---

## [Decision Letter · Decision Letter 1]

7 Apr 2020

Influence of transient pressure changes on speech intelligibility: implications for next-generation train travel

PONE-D-19-26567R1

Dear Dr. Rooney,

We are pleased to inform you that your manuscript has been judged scientifically suitable for publication and will be formally accepted for publication once it complies with all outstanding technical requirements.

With kind regards,

Qiang Zeng, Ph.D.

Academic Editor

PLOS ONE

Additional Editor Comments (optional):

Reviewers' comments:

Reviewer's Responses to Questions

**Comments to the Author**

1. If the authors have adequately addressed your comments raised in a previous round of review and you feel that this manuscript is now acceptable for publication, you may indicate that here to bypass the “Comments to the Author” section, enter your conflict of interest statement in the “Confidential to Editor” section, and submit your "Accept" recommendation.

Reviewer #1: All comments have been addressed

Reviewer #2: All comments have been addressed

2. Is the manuscript technically sound, and do the data support the conclusions?

Reviewer #1: Yes

Reviewer #2: Yes

3. Has the statistical analysis been performed appropriately and rigorously? 

Reviewer #1: Yes

Reviewer #2: Yes

4. Have the authors made all data underlying the findings in their manuscript fully available?

Reviewer #1: No

Reviewer #2: Yes

5. Is the manuscript presented in an intelligible fashion and written in standard English?

Reviewer #1: Yes

Reviewer #2: Yes

6. Review Comments to the Author

Reviewer #1: This study is to investigate the effect of transient change of atmospheric pressure on speech intelligibility in travel train. It seems to be a good complement to the two comments previously pointed out. The reason for selecting SNR as 0 dB and the explanation for the experimental setup seems to be well complemented.

In particular, the originality of this study is very high because studies on the effect of environmental factors such as ambient pressure on speech intelligibility have been tried very little, so it is considered that this paper is suitable for publication in PLOS ONE.

Reviewer #2: The authors followed all my suggestions, and my comment is now positive.

The paper is now ready for being published

7. PLOS authors have the option to publish the peer review history of their article (what does this mean?). If published, this will include your full peer review and any attached files.

Reviewer #1: No

Reviewer #2: No

---

## [Editor Report · Acceptance letter]

9 Apr 2020

PONE-D-19-26567R1 

Influence of transient pressure changes on speech intelligibility: implications for next-generation train travel 

Dear Dr. Rooney:

I am pleased to inform you that your manuscript has been deemed suitable for publication in PLOS ONE. Congratulations! Your manuscript is now with our production department. 

With kind regards,

on behalf of

Dr. Qiang Zeng 

Academic Editor

PLOS ONE